# Cultivated St. John’s Wort Flower Heads Accumulate Tocotrienols over Tocopherols, Regardless of the Year of the Plant

**DOI:** 10.3390/plants14060852

**Published:** 2025-03-09

**Authors:** Ieva Miķelsone, Elise Sipeniece, Inga Mišina, Elvita Bondarenko, Paweł Górnaś

**Affiliations:** Institute of Horticulture, Graudu 1, LV-3701 Dobele, Latvia; ieva.mikelsone@lathort.lv (I.M.); elise.sipeniece@llu.lv (E.S.); inga.misina@llu.lv (I.M.); elvita.bondarenko@llu.lv (E.B.)

**Keywords:** Hypericaceae, herb, aerial part, tocol, α-tocopherol, phytochemicals, lipophilic bioactive compound

## Abstract

St. John’s wort (*Hypericum perforatum* L.) has been extensively utilized across various traditional medicinal systems, including ancient Greek medicine, traditional Chinese medicine, and Islamic medicine. *H. perforatum* is a well-known medicinal plant due to the presence of hypericin and hyperforin, which are natural antidepressants. Recent studies indicate that the inflorescences of wild *H. perforatum* are a source of rare tocotrienols, primarily δ-T3. Similar studies are lacking for cultivated species. *H. perforatum* was grown for three years. At full bloom each year, the plant was cut and separated into its parts: stems, leaves, flower buds, and flowers. Tocotrienols (T3s) were present in each part of the *H. perforatum*. The lowest concentration of tocotrienols was recorded in stems and the highest in flower buds (1.7–4.2 and 88.2–104.7 mg/100 g dry weight, respectively). Flower buds and flowers were the main source of α-T3 and δ-T3 tocotrienols. The plant part has a significant impact on the tocochromanol profile and concentration, while the year of harvest/plant aging does not. The present study demonstrates that cultivated *H. perforatum* flower heads are the first known flowers with relatively high concentrations of tocotrienols. St. John’s wort flower buds accumulate tocotrienols over tocopherols, regardless of the year of the plant.

## 1. Introduction

The genus *Hypericum* belongs to the Hypericaceae family, which is used interchangeably with the Clusiaceae family (Hypericaceae = Clusiaceae) according to the Angiosperm Phylogeny Group (APG) system (1998 version) [1]. Since 2003, the APG system has recognized Hypericaceae Juss. (1789) and Clusiaceae Lindl. (1836) as two separate families [2]. Genus *Clusia* (Clusiaceae) and *Hypericum* (Hypericaceae) belonged to the Clusiaceae family up until 2003 according to the APG system. The chemistries of those two genera seem to have some similarities; for instance, the presence of relatively high concentrations of tocotrienols (lipophilic antioxidants) in their leaves [3,4]. Tocotrienols are generally not found in the leaves of higher plants [5]. While the *Hypericum* genus is well characterized regarding secondary metabolites such as acylphloroglucinols, naphthodianthrones, and polyphenols [6], knowledge on lipophilic secondary metabolites like tocochromanols and carotenoids remains limited. Recent studies indicate that *H. perforatum* inflorescences are a relatively rich source of tocotrienols, particularly δ-T3 [7]. A high tocotrienol content is uncommon in nature, especially in photosynthetic tissues, and the dominance of the δ homologue from four tocotrienols outside of the *Clusia* genus and *H. perforatum* [3] is not widely known. The most recognized source of δ-T3 is annatto (*Bixa orellana*) seeds grown in tropical regions [7], whereas *H. perforatum* is widely distributed in temperate climate zones. Investigating the presence of tocotrienols in plants is important from a health perspective due to evidence from preclinical and clinical studies regarding the therapeutic potential of tocotrienols in inflammation and chronic diseases [8]. Given the rarity of tocotrienols in nature and the widespread geographical distribution of *H. perforatum*, further exploration of this medicinal plant, specifically regarding its lipophilic secondary metabolites, is warranted.

In the photosynthetic tissues of higher plants, α-T is the predominant tocochromanol, whereas tocotrienols are either present in markedly lower concentrations (trace amounts) or entirely absent [5]. Tocochromanols are lipophilic prenyllipid antioxidants, the most common forms being tocopherols (Ts) and tocotrienols (T3s) (Figure 1). 

α-T is a lipophilic antioxidant, one of the main tocochromanols in the leaves of different species. Its main role in plants is to protect photosystem II against photodamage under fluctuating external conditions (environmental stress). In tomatoes, an RNAi line (vte5) characterized by suppressed VTE5 expression and diminished α-T content, the plant’s defensive response to combined high-light and high-temperature stress is through the modulation of α-T production in tomato leaves [9]. The presence of tocotrienols in leaves is rarely documented in the literature; a notable exception is the monocot species *Vellozia gigantea*, for which tocotrienol concentrations were found to be approximately 2.5 times lower than those of tocopherols. The levels of both tocopherols and tocotrienols in the leaves of *V. gigantea* were influenced by plant size and seasonal variations [10]. The tocotrienol content in leaves can be modified (increased) to become dominant over tocopherols through the application of transgenic engineering techniques, as has been demonstrated in studies involving tobacco (*Nicotiana tabacum*) [11]. Despite the widespread adoption of genetically modified (GM) crops by agricultural producers across numerous countries and their prevalence in international food and feed markets, consumer acceptance remains considerably low [12]. Given that natural sources are generally more accepted by consumers than transgenic sources, *H. perforatum* warrants increased attention as a natural source of relatively high levels of tocotrienols, particularly in its inflorescences [7]. Consumer perception influences the market, and natural extracts are generally seen as preferable to genetically modified sources.

*H. perforatum* is the most investigated species in the genus *Hypericum*. The cultivation of St. John’s wort has health, economic, social, and aesthetic aspects. Carefully planning the cultivation of *Hypericum* seems to be a key factor in achieving high and steady biomass yields and the highest content of metabolites [13,14,15]. Both the biomass and phytochemical response of *H. perforatum* are affected by the growth conditions, environmental factors (e.g., water and temperature stress), plant aging, and genotypes [13,14]. The factors described above have not been studied in the context of tocochromanols. Previous studies have asserted that cultivation year and growth conditions significantly influence the phytochemical composition, especially hypericin levels, which is one of the primary reasons *H. perforatum* plants are cultivated on an industrial scale [13]. Currently, there is limited research on the impact of agronomic conditions on *Hypericum* species, and existing studies predominantly examine St. John’s wort extracts, derived from the entire plant or top aerial part of the plant, as sources of bioactive compounds [7,13], without specifically focusing on individual plant parts like stems, leaves, flowers and flower buds.

The aforementioned observations highlight the importance of both plant developmental stage and age on the content of secondary metabolites. Previous studies have concentrated on wild *H. perforatum* [16,17], whereas there is a dearth of research on the cultivated variety, especially concerning long-term observations. Examining the cultivated species is significant not only from a genetic standpoint but also due to the surrounding plant communities accompanying wild St. John’s wort, which could influence the tocochromanol content in this medical plant [17]. Further research constitutes a crucial step toward a better understanding of the biosynthesis of these rare metabolites, specifically tocotrienols, in *H. perforatum*. Therefore, to fully use the potential of St. John’s wort, which, in our estimation, is currently undervalued, further investigation is warranted. Consequently, this study aims to profile tocochromanols, including tocotrienols, in different parts (stems, leaves, flower heads–opened flowers and flower buds) of cultivated *H. perforatum* over three years.

## 2. Results and Discussion

### 2.1. Tocopherols and Tocotrienols in Four Aerial Parts of Cultivated H. perforatum

An analysis of the tocochromanols using reversed-phase (RP) high-performance liquid chromatography (HPLC) with fluorescence detection (FLD) revealed the presence of four tocopherol (T) and four tocotrienol (T3) homologues (α, β, γ, and δ), at least in trace amounts, in all four aerial parts of the analyzed cultivated *H. perforatum* (Table 1). The identification of tocochromanols in St. John’s wort using RP-HPLC-FLD was confirmed via high-resolution mass spectrometry (HRMS) using atmospheric pressure chemical ionization (APCI) to eliminate potential misidentification by the FLD.

The minimum, maximum, average, standard deviation (STDEV), coefficient of variation, the sum of total tocopherols, tocotrienols, and tocochromanols, and the ratio of Ts/T3s for the individual parts of the cultivated *H. perforatum* determined during 2022–2024 are shown in Table 1. α-T was the dominant tocopherol homologue in all the analyzed aerial parts of *H. perforatum*. The other tocopherol homologues (β, γ, and δ) were noted with much lower or negligible amounts. The highest concentrations of α-T were recorded in the leaves. The highest maximum values for the remaining seven tocochromanols were recorded in the flower buds. This, in turn, contributed to the highest concentrations of total tocopherol content being recorded in the leaves, while the flower buds exhibited the highest concentrations of total tocotrienol and tocochromanol content. The stems exhibited the lowest concentrations of both tocopherols and tocotrienols. Concerning tocotrienols, the δ homologue was dominant in the stems, leaves, and flowers, while the α homologue was dominant in the flower buds (Table 1). Tocopherols are distributed across both subterranean (tubers and roots) and aerial parts (seeds, fruits, flowers, flower buds, leaves, and stems) of higher plants. Generally, various plant tissues exhibit a predominance of α-T; however, seeds may contain either α-T or γ-T as the principal tocopherol variant [5]. α-T in photosynthetic tissues contributes to protection mechanisms and serves as the plant’s first line of defense, particularly in protecting the photosynthetic apparatus [18]. The green tissues of higher plants are generally characterized by a notably low concentration of tocotrienols, with reports indicating either a complete absence or only negligible amounts of these compounds. This observation highlights the limited distribution of tocotrienols in the vegetative parts of plants [5]. Recent studies suggest that the presence of relatively high quantities of tocotrienols in leaves appears to be a distinguishing characteristic of both the *Hypericum* and *Clusia* genera [3,4]. Both current and previous studies [3] have demonstrated that δ-T3 is a characteristic tocotrienol homologue in the leaves of *H. perforatum*. This is a rather unique observation, as α and γ are the most prevalent tocotrienol homologues in other *Hypericum* species [3,4] as well as in different species of seeds and their oils [19].

### 2.2. Distribution of Tocochromanols in Cultivated H. perforatum: Flower Heads in the Spotlight

Tocopherols dominated in the leaves (91%, 88% of α-T) and stems (62%, 56% of α-T), but tocotrienols dominated in the flower buds (60%, mainly two: 33% of α-T3 and 18% of δ-T3), while in the flowers, their ratio was relatively balanced (53% of tocopherols and 47% of tocotrienols). It is worth noting that δ-T3 in the stems and leaves constituted 35% and 8% of the total tocochromanol content, respectively. Conversely, in the flowers, δ-T3 predominated over α-T3 (25% and 15%, respectively), in contrast to the flower buds (Figure 2).

The search for information on the tocochromanol content in various flower species yielded fourteen reports examining sixteen different plant species. None of these reports indicated a predominance of tocotrienols over tocopherols. Moreover, with the exception of two instances in which β-T in *Amaranthus caudatus* [20] and δ-T in *Juglans regia* [21] were the dominant homologues in flowers, α-T is the dominant homologue in species such as *Aloe vera* [22], *Borago officinalis* and *Centaurea cyanus* [23], *B. officinalis*, *Camellia japonica*, *C. cyanus* and *Viola × wittrockiana* [24], *Capparis spinosa* [25], *C. cyanus* [26], *Lilium* spp. [27,28], *Moringa oleifera* [29], *Narcissus poeticus* [30], *Tagetes* spp. [31] *Urtica leptophylla* [32], and the edible petals of *Calendula officinalis, C. cyanus*, *Dahlia mignon*, and *Rosa damascena* [33]. Subsequently, γ-T was most frequently the second most abundant homologue after α-T, with its total quantity constituting an average of 10–20% of the total tocopherol content in flowers. In the aforementioned fourteen reports on tocochromanol profiles and content in flowers, only four utilized tocotrienol standards. This suggests that the discovery potential of new tocotrienol sources is significant and warrants future research employing tocotrienol standards. Three-year studies of wild *H. perforatum* populations in Poland [16] and Latvia [17] have shown similar observations, namely the predominance of tocotrienols in flower heads, as in the present study. One important difference noted between wild and cultivated populations is that, regardless of location and year, the flower buds of wild St. John’s wort exhibited a higher content of δ-T3 than α-T3 [16,17], while the cultivated populations showed the opposite trend. Current studies and previous reports do not provide a basis for explaining this rather significant difference between cultivated and wild St. John’s wort. This uniqueness necessitates further investigation. Currently, *H. perforatum* flower heads of both cultivated and wild populations are the undisputed and sole source of tocotrienols in this part of the plant.

### 2.3. Impact of Harvest Year and Plant Part on Tocopherols and Tocotrienols Content in Cultivated H. perforatum

The applied statistical analysis aimed to evaluate the effects of two factors— ‘year’ and ‘plant part’—on eight dependent variables—’α-T’, ‘β-T’, ‘γ-T’, ‘δ-T’, ‘α-T3’, ‘β-T3’, ‘γ-T3’, and ‘δ-T3’. The results are presented as boxplots (showing medians) to illustrate the distribution of dependent variables across groups defined by the ‘year’ and ‘plant part’ factors. In the case of both tocopherols and tocotrienols, the year of harvest (2022–2024)/plant age (1–3) did not have a statistically significant effect on the content of these lipophilic secondary metabolites (Figure 3 and Figure 4). Previous studies in wild *H. perforatum* populations [16,17], similar to the present study, have shown a negligible effect of the harvest year on the content of tocopherols and tocotrienols in the plant. The lack of significant differences across different harvest years is good news for *H. perforatum* growers, as it allows for a stable estimation of the economic benefits derived from the dual utilization of St. John’s wort (primarily as a source of hypericin and hyperforin, as well as tocotrienols). Opposite to our findings, in studies analyzing eight secondary metabolites, including hyperforin, hypericin, and several flavanols, it was observed that their quantities were statistically greater in the second year of harvest compared to the first. Additionally, it was noted that the influence of genetics was more significant than environmental effects [14]. Similar observations regarding the concentration of hyperforin, hypericin, and flavanols have been reported in four-year studies, in which substantial differences were noted between the studied years and the age of the plants. Furthermore, local genotypes were generally found to be more suitable for field cultivation [13]. In the current study, the plant material (seeds) originated from a collection at the botanical garden in Wrocław, Poland. The plant performed exceptionally well under Latvian conditions. The current research suggests two key points: first, plant age and environmental conditions have a varied impact on the content of different secondary metabolites (tocotrienols vs. hyperforin and hypericin); second, the functions of the generated secondary metabolites in *H. perforatum* seem to be different. Understanding these relationships and functions requires further investigation.

The effect of the plant part was significant, but that of the year was not. In most cases, the aerial parts of the plant had a statistically significant impact (*p* < 0.005) on the content of tocochromanols, with a few exceptions. No significant differences were observed for two pairs: flowers and flower buds (five instances), as well as leaves and stems (three instances). For the pair of flowers and flower buds, statistically significant differences (*p* < 0.005) were recorded only for the content of three tocotrienols (α-T3, β-T3, and γ-T3). Conversely, for the pair of leaves and stems, no significant differences were noted for δ-T, β-T3, and γ-T3. The predominance of tocotrienols in the flower heads of cultivated St. John’s wort was indisputable, and their content was higher in the flower heads than in the leaves and stems. Similar observations were previously reported in wild populations of St. John’s wort [16,17]. In the flowers and flower buds, it is noteworthy that there were approximately two to three times higher values of three tocotrienols (α-T3, β-T3, and γ-T3) in the flower buds compared to in the flowers. Significantly lower levels of these three tocotrienols in flowers relative to flower buds have been previously reported in wild St. John’s wort [16,17]. The substantial loss of tocotrienols during the opening of flower buds may be attributed to the exposure of the plant’s reproductive organs to external environmental conditions, a mechanism known as photoinhibition and photoprotection. Such a mechanism has been observed during flower opening in *Lilium*, when an initial increase in tocochromanols during bud development (stages I and II) is followed by a decrease at full opening (stage III), particularly for α-T3. Tocochromanols increased during early development stages, contrasting with other compounds such as carotenoids and xanthophylls, whose content decreased during flower development. A study on *Lilium* flowers revealed that during stage II of development (when the flower is closed, its green color starts to disappear, and it is just about to open after 5 days of monitoring), both tocopherol and tocotrienol concentrations increased, with α-T being the predominant tocochromanol, followed by α-T3 and γ-T. The study also demonstrated that a plant treatment agent containing hormones reduced tocochromanols [28]. Similar observations between St. John’s wort (dicots) and *Lilium* (monocots) may suggest a comparable role for α-T3 in flowers regardless of species or classification; however, additional studies are required to confirm this observation and more precisely explain the differences between flower buds and flowers. This may necessitate more detailed investigations into flowers by dissecting them into individual parts (stamen, pistil, petals, and sepals).

## 3. Materials and Methods

### 3.1. Reagents

The following chemicals were used as received: ethanol, ethyl acetate, methanol, and *n*-hexane (HPLC grade) from Sigma-Aldrich (Steinheim, Germany), alongside pyrogallol, sodium chloride, and potassium hydroxide (reagent grade) from the same supplier. Tocopherol and tocotrienol homologue standards (α, β, γ, and δ) with a purity exceeding 98% (HPLC) were obtained from Extrasynthese (Genay, France) and Cayman Chemical (Ann Arbor, MI, USA).

### 3.2. Plant Material

The seeds of cultivated *H. perforatum* were procured from the Botanical Garden of Medicinal Plants, affiliated with the Department of Biology and Pharmaceutical Botany at Wrocław University of Medicine, located in Wrocław, Poland. Seeds were sown in small peat containers in the beginning of March 2022 and cultivated for four weeks. Then, the strongest plants were picked up and transplanted into individual container cells (5 × 5 cm). The plants were kept in a greenhouse without additional light or heat conditions and watered to prevent them from drought stress. Peat KKS-U parameters according to manufacturer LaFlora (Līvbērzes, Latvia) were as follows: pH/KCl 5.2–6.0; grind 0–40 mm; PG-Mix 15–10–20 1.0 kg/m^3^; Floraspur 100 g/m^3^; Osmocot 1 kg/m^3^; clay 8 kg/m^3^; and wetting agent 0.3 L/m^3^. Sixty of the healthiest St. John’s wort plants were planted in the open field, on the first days of June 2022, located in the garden of the Institute of Horticulture, Dobele, Latvia (GPS location: N: 56°36′39″ E: 23°17′50″). In the field, plants were planted in specially designed beds covered with agrofilm and provided with water irrigation during long periods of hot and dry weather. The soil parameters were as follows: pH 7.6; organic matter 2%; Mg—1510 mg/kg; Ca—3592 mg/kg; Fe—480 mg/kg; N-NO_3_—27 mg/kg; and N-NH_4_—1 mg/kg (according to State Plant Protection Service of the Republic of Latvia analysis). St. John’s wort was harvested in 2022−2024 in full bloom (Appendix A) by cutting ten plants 5–10 cm from the soil and separated into four aerial parts (stems, leaves, flower buds, and flowers) for each of the three biological replicates. Separate parts were freeze-dried using a FreeZone freeze dry system (Labconco, Kansas City, MO, USA) at a temperature of −51 ± 1 °C under vacuum of below 0.01 mbar for 48 h. The plant material obtained after freeze drying, which constituted 3−10 g for each sample, was transferred into polypropylene tubes and stored at −18 ± 1 °C until milling (not longer than 1 month before the analysis). The dry plant material was powdered using an MM 400 mixer mill (Retsch, Haan, Germany) and stored at −18 ± 1 °C until its use (not longer than 1 month). The dry mass was measured gravimetrically.

### 3.3. H. perforatum Sample Preparation for Tocochormanols’ Determination (Saponification Protocol)

For tocochromanols’ extraction from all *H. perforatum* parts, the most frequently used protocol, saponification, was applied due to it having the highest recovery of these prenyllipids. After saponification, the samples were extracted three times with a *n*-hexane:ethyl acetate mixture (9:1, *v*/*v*), evaporated to dryness, and then reconstituted in ethanol. The entire procedure was performed according to a previously established protocol [34].

### 3.4. Tocochromanols’ (Tocopherols and Tocotrienols) Determination by RP-HPLC-FLD

The tocochromanol analyses were performed using reversed-phase high-performance liquid chromatography with fluorescent light detector (RP-HPLC-FLD) via HPLC Shimadzu Nexera 40 Series system (Kyoto, Japan) consisting of a pump (LC-40D pump), a degasser (DGU-405), a system controller (CBM-40), an auto injector (SIL-40C), a column oven (CTO-40C), and a fluorescence detector (RF-20Axs). The chromatographic separation of tocopherol and tocotrienol homologues was carried out on an Epic PFP-LB (pentafluorophenyl phase) column (PerkinElmer, Waltham, MA, USA) with the following parameters: particle morphology—fully porous; particle size—3 µm; column length—150 mm; column ID—4.6 mm; secured with a 4 mm long guard column; and ID—3 mm (Phenomenex, Torrance, CA, USA). The chromatography analysis was performed under the isocratic conditions: mobile phase—methanol with water (91:9; *v*/*v*); flow rate—1.0 mL/min; column oven temperature—45 ± 1 °C; and room temperature—21 ± 1 °C. The total chromatography runtime was 13 min. The retention times for individual tocochromanols were as follows: 5.3 min—δ-T3, 6.1 min—β-T3, 6.5 min—γ-T3, 7.3 min—α-T3, 7.9 min—δ-T, 9.3 min—β-T, 9.8 min—γ-T, and 11.4 min—α-T. The identification and quantification were performed using a fluorescence detector at an excitation wavelength of 295 nm and emission wavelength of 330 nm. The quantification was performed based on the calibration curves obtained from tocopherol and tocotrienol standards. The detailed parameters of the method were previously developed and validated to ensure accuracy and reliability [35]. Representative RP-HPLC-FLD chromatograms of the tocopherol (T) and tocotrienol (T3) homologues’ (α, β, γ, and δ) separation in cultivated *H. perforatum* stems, leaves, inflorescences (flower buds and flowers), and standards are illustrated in Appendix A.

### 3.5. RPLC-APCI-HRMS Analysis

The obtained results of tocotrienols’ and tocopherols’ identification and quantification via RP-HPLC-FLD were confirmed by liquid chromatography–atmospheric pressure chemical ionization–high-resolution mass spectrometry (RPLC-APCI-HRMS). All parameter details of this analysis have been described previously [36]. In summary, the analytical workflow was conducted using a Q-Exactive Orbitrap MS system (Thermo Scientific, Dreieich, Germany) coupled with an Ultimate 3000 HPLC system (Dionex, Sunnyvale, CA, USA). The separation was achieved on a Kinetex PFP column (1.7 µm, 100 × 3 mm; Phenomenex, Torrance, CA, USA) with a binary mobile-phase system of water (A) and methanol (B). The gradient elution protocol included 20% B for 1 min, a linear increase from 20% to 95% B between 1 and 9.5 min, a hold at 95% B from 9.5 to 25 min, and a return to 20% B from 25.1 to 28 min. The flow rate was set at 0.3 mL/min. Tocochromanols were quantified in full scan mode at a resolution of 70,000 FWHM (at 200 *m*/*z*) over a range of 100 to 1000 *m*/*z*, using negative atmospheric pressure chemical ionization (APCI) mode. The identification of compounds was based on comparing peak areas of corresponding deprotonated [M-H]⁻ ions, with a mass accuracy tolerance of ± 5 ppm and a retention time tolerance of ±0.1 min. The LOQ was determined by analyzing standard solutions at low concentrations (0.01 to 0.25 ng/μL), with the lowest concentration producing an S/N ratio of ≥10 designated as the LOQ. Data acquisition and processing were carried out using Thermo Scientific Xcalibur software (v. 4.1).

### 3.6. Statistical Analysis

Results presented in the form of table, referring to minimum, maximum, and average content with standard deviation and coefficient of variation and content proportion (%), are based on three independent biological replications of *H. perforatum* during three-year study (*n* = 9 = 3 × 3) for each plant part (stems, leaves, flower buds, and flowers). The results were visualized with the assistance of Excel (Version 2302) Microsoft 365 Apps for Enterprise (Redmond, WA, USA) software.

The statistical analysis aimed to evaluate the effects of two factors—‘year’ and ‘plant part’—on eight dependent variables—’α-T’, ‘β-T’, ‘γ-T’, ‘δ-T’, ‘α-T3’, ‘β-T3’, ‘γ-T3’, and ‘δ-T3’. Given the non-normal distribution of the data, statistical methods appropriate for non-normal and heteroscedastic data were employed. Specifically, the Scheirer–Ray–Hare test, a non-parametric analog of two-way analysis of variance (ANOVA), was used to independently assess the influence of each factor on the dependent variables, without requiring assumptions of normality. This test was conducted separately for each dependent variable, and test statistics and *p*-values were calculated for both factors (‘year’ and ‘plant part’). For dependent variables exhibiting significant effects in the Scheirer–Ray–Hare test (*p* < 0.05), post-hoc pairwise comparisons were performed using the non-parametric Mann–Whitney U test to compare two independent groups. The results were corrected for multiple comparisons using the Bonferroni method to limit the risk of Type I error. The results are presented as boxplots (showing medians) to illustrate the distribution of dependent variables across groups defined by the ‘year’ and ‘plant part’ factors. These visualizations facilitated the identification of patterns and significant differences between groups. Statistical analyses were conducted using the Python programming language (3.12.7 packaged by Anaconda, Inc., Austin, TX, USA) with the following libraries: ‘scipy.stats` for statistical tests (Scheirer–Ray–Hare and Mann–Whitney U); ‘matplotlib’ and ‘seaborn’ for data visualization; and ‘pandas’ for data processing and analysis.

## 4. Conclusions

This study demonstrated that cultivated St. John’s wort flower heads accumulate tocotrienols, mainly α-T3 and δ-T3, over tocopherols, regardless of the year of the plant. Flower buds contain 60% tocotrienols, and flowers contain 47% tocotrienols, which means that if flower buds constitute at least 24% and the remaining 76% are flowers, then tocotrienols will account for over 50% of the tocochromanols in St. John’s wort flower heads. This is a relevant discovery further emphasizing the uniqueness of *H. perforatum* regarding its unique secondary metabolites exhibiting health-promoting properties. This finding has the potential to have important implications not only for plant science but also for the medical/pharmaceutical sector, which mainly uses hydrophilic bioactive compounds from St. John’s wort.

The plant part had an important impact on the tocochromanol profile and concentration, while the year of harvest did not. The rather high decrease in α-T3 content between flower buds and flowers requires future consideration to better understand the role of tocotrienols in *H. perforatum* flowers. It is also important to explain the unclear difference in the predominance of α-T3 in the flower buds of cultivated St. John’s wort and δ-T3 in wild St. John’s wort.

As the average content of tocotrienols, mainly α-T3, is two times higher in flower buds, it is suggested that St. John’s wort plants be harvested before reaching full bloom to preserve these valuable lipophilic phytochemicals. To explain the differences between wild populations and cultivated St. John’s wort, as well as the significant loss of tocotrienols at full bloom, further studies are required.

## Figures and Tables

**Figure 1 plants-14-00852-f001:**
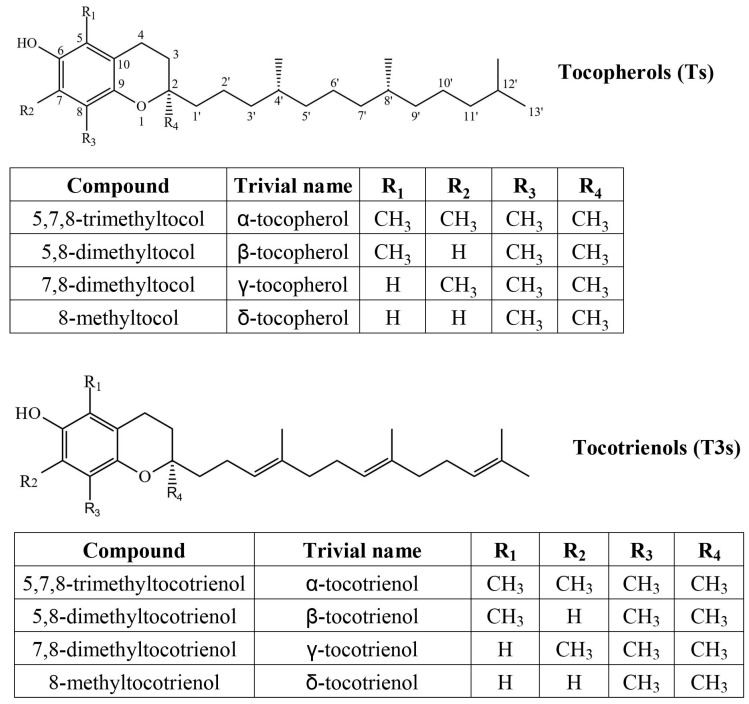
Chemical structures of four tocopherol (T) and four tocotrienol (T3) homologues.

**Figure 2 plants-14-00852-f002:**
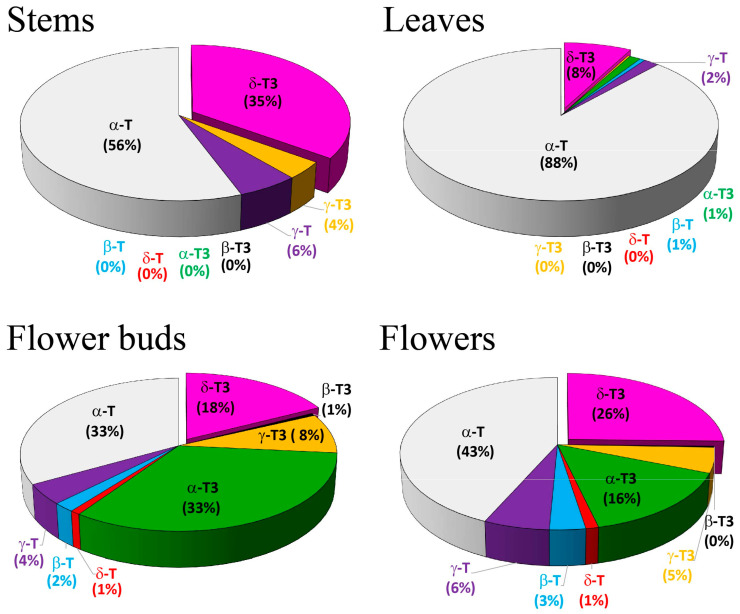
The average proportion (%) of individual tocotrienol (T3) and tocopherol (T) homologues (α, β, γ, and δ) in stems, leaves, flower buds, and flowers of cultivated *H. perforatum* harvested during 2022–2024.

**Figure 3 plants-14-00852-f003:**
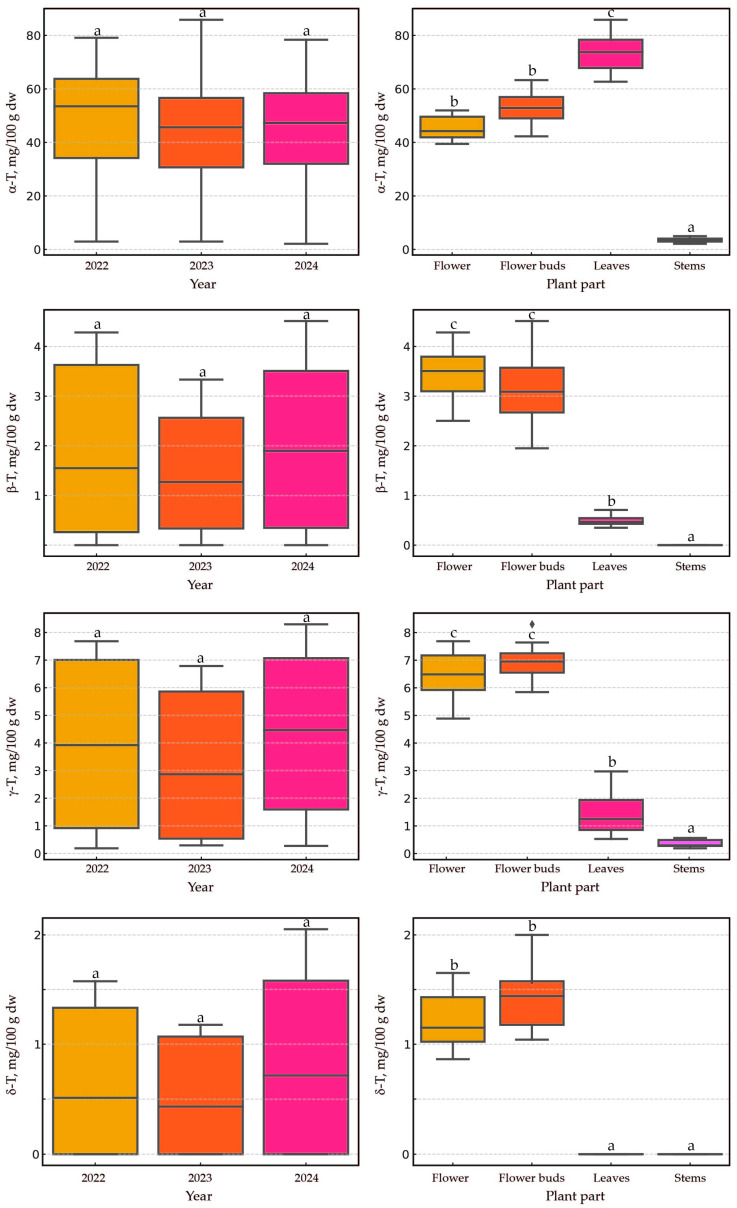
The boxplots illustrate the distribution of four dependent variables (α-T, β-T, γ-T, and δ-T) in cultivated *H. perforatum* across groups defined by the ‘year’ and ‘plant part’ factors. Different letters indicate statistically significant differences at *p* < 0.005. T, tocopherol; dw, dry weight.

**Figure 4 plants-14-00852-f004:**
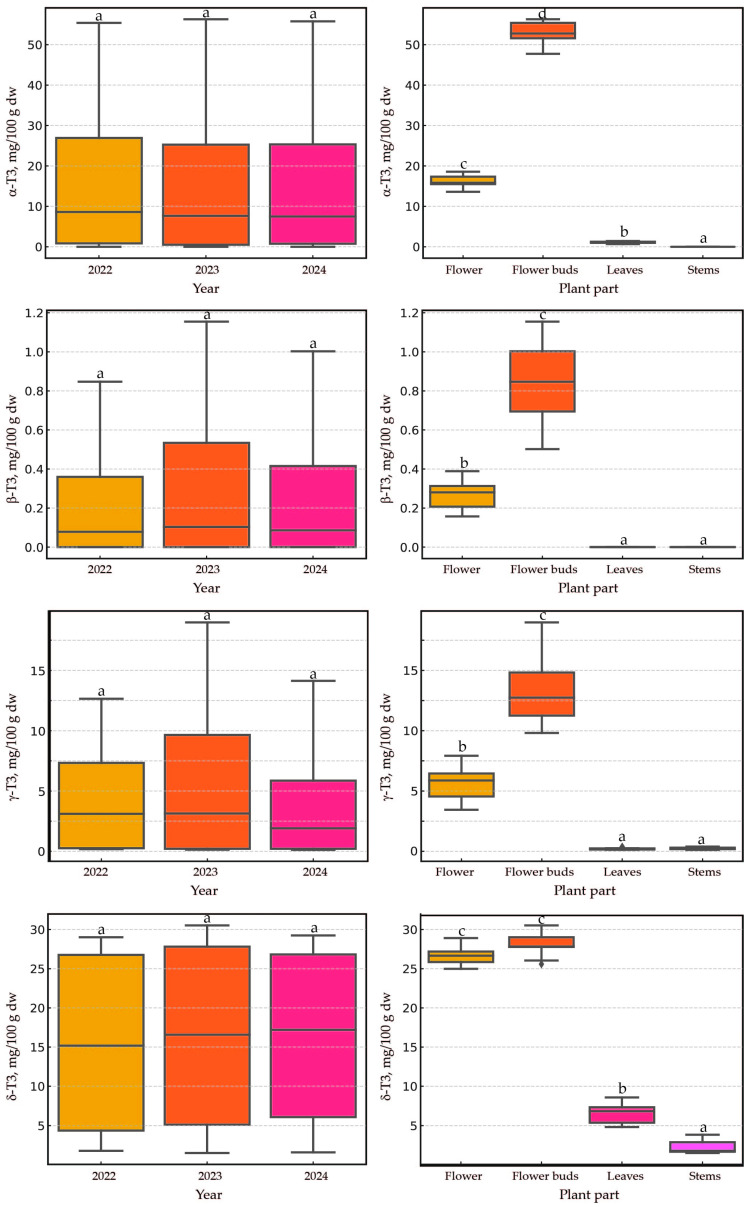
The boxplots illustrate the distribution of four dependent variables (α-T3, β-3T, γ-T3, and δ-T3) in cultivated *H. perforatum* across groups defined by the ‘year’ and ‘plant part’ factors. Different letters indicate statistically significant differences at *p* < 0.005. T3, tocotrienol; dw, dry weight.

**Table 1 plants-14-00852-t001:** Content and ratio of tocopherols and tocotrienols in stems, leaves, flower buds, and flowers of cultivated *H. perforatum* harvested during 2022–2024.

Plant Part	Tocochromanols, mg/100 g dw											Ratio Ts/T3s
	α-T *	β-T	γ-T	δ-T	α-T3	β-T3	γ-T3	δ-T3	Total Ts	Total T3s	Total Ts + T3s	
Stems												
Minimal	2.1	nd	0.2	nd	nd	nd	0.1	1.5	2.4	1.7	4.1	0.9
Maximal	5.0	tr	0.6	tr	tr	tr	0.4	3.8	5.5	4.2	8.2	2.9
Average	3.6	–	0.4	–	–	–	0.2	2.2	3.9	2.5	6.4	1.7
STDEV	0.9	–	0.1	–	–	–	0.1	0.8	1.0	0.9	1.5	0.6
Coefficient of variation	24.5	–	39.0	–	–	–	40.6	37.1	24.6	37.0	23.4	35.4
Leaves												
Minimal	62.7	0.3	0.5	nd	0.7	nd	0.1	4.8	66.4	6.3	73.1	6.6
Maximal	85.8	0.7	3.0	tr	1.4	tr	0.4	8.6	87.3	10.1	95.6	11.9
Average	73.1	0.5	1.5	–	1.1	–	0.2	6.5	75.1	7.8	82.9	9.8
STDEV	7.5	0.1	0.8	–	0.3	–	0.1	1.3	7.2	1.2	7.4	1.5
Coefficient of variation	10.3	23.1	55.7	–	24.5	–	35.6	19.3	9.6	15.3	8.9	15.5
Flower buds												
Minimal	42.3	1.9	5.8	1.0	47.7	0.5	9.8	25.6	53.0	88.2	147.2	0.5
Maximal	63.3	4.5	8.3	2.1	56.3	1.7	19.0	30.5	76.0	104.7	173.9	0.8
Average	52.8	3.1	7.0	1.4	53.0	0.9	13.4	28.0	64.3	95.2	159.6	0.7
STDEV	7.0	0.8	0.7	0.3	2.8	0.2	3.0	1.5	7.4	5.9	8.4	0.1
Coefficient of variation	13.3	25.7	10.5	22.1	5.4	25.5	22.4	5.5	11.5	6.2	5.3	13.5
Flowers												
Minimal	39.4	2.5	4.9	0.9	13.6	0.2	3.4	25.0	48.9	43.1	97.1	1.0
Maximal	52.0	4.3	7.7	1.7	18.6	0.4	7.9	29.0	65.0	52.0	117.0	1.3
Average	45.6	3.5	6.4	1.2	16.1	0.3	5.7	26.8	56.8	48.8	105.6	1.2
STDEV	4.6	0.6	0.9	0.3	1.6	0.1	1.5	1.2	5.3	2.5	6.2	0.1
Coefficient of variation	10.0	16.2	14.7	21.8	10.0	29.8	26.6	4.4	9.3	5.1	5.8	9.8

* Average values and standard deviations correspond to nine biological samples of each aerial part of *H. perforatum* (*n* = 9). T, tocopherol; T3, tocotrienol; tr, trace amount (below 0.05 mg/100 g dw); dw, dry weight.

## Data Availability

Data are contained within the article and Appendix A.

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
