# Peer review of "Cultivated St. John’s Wort Flower Heads Accumulate Tocotrienols over Tocopherols, Regardless of the Year of the Plant"

_plants, 2025, doi:10.3390/plants14060852_

Round 1
Reviewer 1 Report
Comments and Suggestions for Authors
Dear authors,
I have several minor remarks.
Could you describe the novelty of your study in the introduction section? This plant species is deeply studied, so it would be good if you explain the novelty.
You should provide reference for the development and validation of the HPLC methods involved in your study. However, if these methods are used for the first time in your study you should provide Validation data.
Secondly, you should provide chromatograms (in supplementary files, or you could include the chromatograms as figures).
The HPLC methods should be better described. Please include the total time for analysis. Moreover, you must include data about the retention time of each compound.
Figure 1.-could you provide better quality figure?
Author Response
We sincerely thank you for all the comments, remarks, and suggestions that have contributed to enhancing the manuscript and its scientific quality. The manuscript and supplementary materials have been improved accordingly. Provided changes are marked in red font. For the literature we used references manager software, therefore the changes are not highlighted.
Reviewer 1
Comment 1: I have several minor remarks.
Could you describe the novelty of your study in the introduction section? This plant species is deeply studied, so it would be good if you explain the novelty.
Response 1: Thank you for your comment. We have updated introduction with explaining the novelty of this study. Please see: page 3, bottom page.
Comment 2: You should provide reference for the development and validation of the HPLC methods involved in your study. However, if these methods are used for the first time in your study you should provide Validation data.
Response 2: Thank you for your comment. Indeed, the method was earlier developed and validated. The appropriate reference was provided. Please see: page 11, bottom page and page 12, top page.
Comment 3: Secondly, you should provide chromatograms (in supplementary files, or you could include the chromatograms as figures).
Response 3: Thank you for your comment. The chromatograms were provided in the Supplementary Materials.
Comment 4: The HPLC methods should be better described. Please include the total time for analysis. Moreover, you must include data about the retention time of each compound.
Response 4: Thank you for your comment. The information about the total time for analysis and retention time of individual tocochromanols was provided. Please see: page 11, bottom page.
Comment 5: Figure 1.-could you provide better quality figure?
Response 5: Thank you for your comment. Well, we think that the quality of the present Figure 2 (earlier Figure 1) is OK. Utilized software was unable to improve it.
Reviewer 2 Report
Comments and Suggestions for Authors
The presented study is a continuation of the previous works of the authors' team, devoted to the study of the accumulation of tocopherols and tocotrienols in the raw material of St. John's wort, which is one of the unique representatives of plants from this point of view. The absence of the influence of the year of plant vegetation on the accumulation of target metabolites is shown, a difference in the accumulation of alpha and beta-tocotrienols in wild and cultivated plants is noted. As a recommendation, in my opinion, this phenomenon should be traced on representatives of other species and the data should be presented in accordance with the newly discovered information, what is common in the habitat of the species accumulating the studied metabolites, what is their role in the producer organisms. A more vivid presentation of these data will allow the reader to better understand the research topic presented by the authors.
Author Response
We sincerely thank you for all the comments, remarks, and suggestions that have contributed to enhancing the manuscript and its scientific quality. The manuscript and supplementary materials have been improved accordingly. Provided changes are marked in red font. For the literature we used references manager software, therefore the changes are not highlighted.
Reviewer 2
Comment 1: The presented study is a continuation of the previous works of the authors' team, devoted to the study of the accumulation of tocopherols and tocotrienols in the raw material of St. John's wort, which is one of the unique representatives of plants from this point of view. The absence of the influence of the year of plant vegetation on the accumulation of target metabolites is shown, a difference in the accumulation of alpha and beta-tocotrienols in wild and cultivated plants is noted. As a recommendation, in my opinion, this phenomenon should be traced on representatives of other species and the data should be presented in accordance with the newly discovered information, what is common in the habitat of the species accumulating the studied metabolites, what is their role in the producer organisms. A more vivid presentation of these data will allow the reader to better understand the research topic presented by the authors.
Response 1: Thank you for the positive overview. Thank you for your comment. Modifications have been introduced to the introduction and discussion sections. Both the current study and prior reports fail to provide a comprehensive understanding of the mechanisms underlying tocotrienol accumulation in H. perforatum. Further detailed investigations are necessary to address this gap. We identified certain similarities in tocochromanols accumulation during the flower formation in Lilium and H. perforatum, which are extensively discussed in the manuscript. Please see: page 3, bottom page and page 10, middle page.
Reviewer 3 Report
Comments and Suggestions for Authors
This very interesting manuscript describes finding tocopherol and tocotrienol derivatives in the St. John's Wort plant with the alpha- and delta-tocotrienol concentration greatest in the flower buds (before full bloom). The concentrations of these natural antioxidant molecules is of interest and they were measured, quantified and identified using chromatographic analytical methods.
The manuscript is written in an easy to read style I have only a few suggested changes:
(1) rethinking where the text about the Lilium species (lines 93-99) and (lines 238-243) goes. It seems to me that placing these two mentions together might help the flow of the manuscript. (2) include a figure or scheme depicting the molecular structures of the 4 tocopherols and the 4 tocotrienols. The slight differences among these 8 molecules can lead to much confusion unless a figure illustrating the chemical structures is included.
Author Response
We sincerely thank you for all the comments, remarks, and suggestions that have contributed to enhancing the manuscript and its scientific quality. The manuscript and supplementary materials have been improved accordingly. Provided changes are marked in red font. For the literature we used references manager software, therefore the changes are not highlighted.
Reviewer 3
Comment 1: This very interesting manuscript describes finding tocopherol and tocotrienol derivatives in the St. John's Wort plant with the alpha- and delta-tocotrienol concentration greatest in the flower buds (before full bloom). The concentrations of these natural antioxidant molecules is of interest and they were measured, quantified and identified using chromatographic analytical methods.The manuscript is written in an easy to read style I have only a few suggested changes:
Response 1: Thank you for the positive overview and your suggestions.
Comment 2: (1) rethinking where the text about the Lilium species (lines 93-99) and (lines 238-243) goes. It seems to me that placing these two mentions together might help the flow of the manuscript.
Response 2: Thank you for your comment. The improvements have been made according to recommendation. Please see: page 10, middle page.
Comment 3: (2) include a figure or scheme depicting the molecular structures of the 4 tocopherols and the 4 tocotrienols. The slight differences among these 8 molecules can lead to much confusion unless a figure illustrating the chemical structures is included.
Response 3: Thank you for your comment. The improvements have been made according to recommendation. Please see: page 2, middle-bottom page.